Serum growth differentiation factor 15 trajectory predicts 28-day mortality in critically ill patients: a multicenter cohort study

Wang Qinxue 1 2
Wang Jiawei 3
Zhao Yuhan 3
Ma Yuanze 3
Li Xiang 1 2
Chang Xinyi 4
Zheng Nan 5
Ji Yong 6 7 yongji@njmu.edu.cn
Han Yi 3 4 yihan@hrbmu.edu.cn
1 Department of Geriatric Intensive Care Unit, First Affiliated Hospital of Nanjing Medical University , Nanjing, Jiangsu , China
2 Department of Critical Care Center, First Affiliated Hospital of Nanjing Medical University , Nanjing, Jiangsu , China
3 Nanjing Medical University , Nanjing, Jiangsu , China
4 Department of Critical Care Medicine, Second Affiliated Hospital of Harbin Medical University , Harbin, Heilongjiang , China
5 Department of Critical Care Medicine, Women’s Hospital of Nanjing Medical University , Nanjing, Jiangsu , China
6 Key Laboratory of Cardiovascular and Cerebrovascular Medicine, Nanjing Medical University , Nanjing, Jiangsu , China
7 State Key Laboratory of Frigid Zone Cardiovascular Diseases, Harbin Medical University , Harbin, Heilongjiang , China
Anson Lesley
Electronic publication date: 2025 Nov 3
Publication date: 2025
Volume: 13
Electronic Location ID: e20317
Received 2025 Mar 25; Accepted 2025 Oct 9
Copyright: © 2025 Wang et al.
Copyright year: 2025
Copyright holder: Wang et al.
License: This is an open access article distributed under the terms of the Creative Commons Attribution License, which permits unrestricted use, distribution, reproduction and adaptation in any medium and for any purpose provided that it is properly attributed. For attribution, the original author(s), title, publication source (PeerJ) and either DOI or URL of the article must be cited.
License URL: https://creativecommons.org/licenses/by/4.0/

Keywords: Growth differentiation factor 15, 28-day mortality, Intensive care unit, Prognostic biomarker, Group-based trajectory modeling

Funding: National Key Research and Development Program of China 2023YFA1801900 National Natural Science Foundation of China 82470496 Outstanding Youth Project of Heilongjiang Natural Science Foundation JQ2024H004 Young Scholars Fostering Fund of the First Affiliated Hospital of Nanjing Medical University PY2023035 The study was supported by grants from the National Key Research and Development Program of China (2023YFA1801900), the National Natural Science Foundation of China (Grant No. 82470496), the Outstanding Youth Project of Heilongjiang Natural Science Foundation (No. JQ2024H004), and the Young Scholars Fostering Fund of the First Affiliated Hospital of Nanjing Medical University (Grant Nos. PY2023035). The funders had no role in study design, data collection and analysis, decision to publish, or preparation of the manuscript.

==============================
Background

Growth differentiation factor 15 (GDF15) has been linked to critical illnesses, particularly cardiovascular and infectious diseases, but its dynamic patterns and prognostic value in critically ill patients remain unclear. This study investigates the predictive utility of serum GDF15 trajectories for 28-day mortality among patients in the intensive care unit (ICU).

Methods

In this multicenter, prospective cohort study, ICU patients were enrolled, and serum GDF15 trajectories during the first week were analyzed using group-based trajectory modeling (GBTM). The association between trajectory subtypes and 28-day mortality was assessed through hierarchically adjusted multivariable logistic regression. A cumulative index, “GDF15-load,” was introduced to quantify overall GDF15 exposure and compared with the Acute Physiology and Chronic Health Evaluation II (APACHE II) and Sequential Organ Failure Assessment (SOFA) scores. The correlation between initial GDF15 levels (GDF15-D1) and GDF15-load was evaluated using Spearman’s correlation test. Predictive performance was assessed via the area under the receiver operating characteristic curve (AUROC), and feature importance was interpreted using Shapley Additive Explanations (SHAP).

Results

Among 1,973 patients, 493 comprised the cohort for development with full serum profiles on days 1, 3, and 7. Four GDF15 trajectory subtypes were identified: low-maintenance (LM), medium-maintenance (MM), high-increase (HI), and high-persistent (HP). Trajectory subtypes showed significant differences in inflammatory markers, organ dysfunction, and 28-day mortality, with the HI and HP groups having the worst outcomes. GDF15-load increased progressively from LM to HP and emerged as the most important predictor of 28-day mortality, not inferior to APACHE II and SOFA scores. GDF15-D1 was strongly correlated with GDF15-load (Spearman r = 0.778) and demonstrated robust predictive value, particularly in postoperative ICU patients, where its combination with APACHE II or SOFA further improved prognostic accuracy.

Conclusions

Serum GDF15 trajectory and GDF15-load are robust predictors of 28-day mortality in ICU patients. GDF15-D1 strongly reflects cumulative GDF15 burden and provides a rapid, practical tool for early risk stratification, especially in postoperative ICU patients. These findings support the use of GDF15 as both a dynamic and point-of-care biomarker in intensive care settings.

Introduction

In critical care practice, prognostic evaluation remains a pivotal concern. Early and accurate identification of critically ill patients at high risk of death enables timely and individualized interventions.

For prognostic assessment in critically ill patients, several scoring systems are available, among which the Acute Physiology and Chronic Health Evaluation II (APACHE II) remains the most widely used (Bion et al., 1988; Kruse, Thill-Baharozian & Carlson, 1988; Quintairos, Pilcher & Salluh, 2023). A retrospective analysis showed that the APACHE II score on day 3 of intensive care unit (ICU) admission offers optimal prognostic value, with a cut-off of 17 to distinguish high-risk patients (Tian et al., 2021). However, as it requires 24-h data collection, there is increasing interest in identifying complementary indicators that enable earlier risk stratification before the full score becomes available.

Serum biomarkers that can be rapidly measured offer valuable tools for early prognostic assessment, whether applied individually or within panels or integrated models (Dieplinger et al., 2016; Leaf et al., 2017; Mikacenic et al., 2017). Among these, growth differentiation factor 15 (GDF15) has attracted growing attention. As a stress-responsive cytokine induced by cellular injury, inflammation, and hypoxia, GDF15 is increasingly recognized for its association with poor outcomes in ICU patients (Unsicker, Spittau & Krieglstein, 2013; Verhamme et al., 2019). Recent studies suggest its potential as a mortality predictor in various critically ill populations, including sepsis, cardiogenic shock, acute respiratory distress syndrome (ARDS), and acute pulmonary embolism (Buendgens et al., 2017; Clark et al., 2013; Hongisto et al., 2019; Lankeit et al., 2008; Meijers et al., 2021).

However, most existing studies have focused on single time-point measurements of GDF15, which may fail to capture the dynamic and evolving nature of critical illness. Similar to how cumulative metrics such as “lactate load” provide enhanced prognostic value, dynamic modeling of GDF15 may offer deeper insights (Chen, Gong & Yu, 2022; Wu et al., 2024). While some studies have linked GDF15 to disease severity, few have explored its longitudinal trajectories or the concept of cumulative exposure, herein referred to as “GDF15-load.” Moreover, whether a single early measurement of GDF15 at ICU admission could serve as a practical surrogate for this cumulative burden remains unclear.

To address these gaps, we conducted a multicenter prospective cohort study to investigate the prognostic utility of GDF15 in critically ill patients. Specifically, we aimed to (1) characterize distinct temporal trajectories of serum GDF15 using group-based trajectory modeling (GBTM); (2) propose and validate “GDF15-load” as a measure of cumulative GDF15 exposure; and (3) evaluate the relationship between initial GDF15 levels at ICU admission and GDF15-load. Through this approach, we sought to establish a dynamic, biologically meaningful, and clinically applicable framework for GDF15 as a prognostic biomarker in critical care.

Materials and Methods

Ethical statement

This study was recorded in the Chinese Clinical Trial Registry (No. ChiCTR2400090770 and No. ChiCTR2400091418) and have been approved by the Institutional Study Board of the First Affiliated Hospital of Nanjing Medical University and the Second Affiliated Hospital of Harbin Medical University (2022-SR-678 and KY2024-066-01). The study was conducted in accordance with the principles of the Helsinki Declaration, and informed consent was obtained from all participants.

Study design and population

This study is a prospective, multicenter cohort conducted from March 2023 to August 2024 across three cities and four ICUs in China. Inclusion criteria included: 1. ≥18 years of age; 2. Enrollment within 24 h of ICU admission (timed from first arrival in the ICU); 3. At least one organ dysfunction, defined as a Sequential Organ Failure Assessment (SOFA) subscore ≥2 in any of the following systems: respiratory, cardiovascular, liver, renal, coagulation, or central nervous system; 4. An expected ICU stay of ≥48 h, as judged by the treating physician. Exclusion criteria included: (1) Expected death or withdrawal of life-sustaining treatment within 48 h after screening; (2) pregnant or breastfeeding; (3) hospitalized for >7 days prior to ICU admission. The study was structured into two phases: the development cohort phase, aiming to enroll 500 patients with a survival period of ≥7 days and serum samples available on days 1, 3, and 7. Following the development cohort, we continuously enrolled a validation cohort of 1,500 patients in two centers, requiring serum samples on at least day 1. The study followed the Strengthening the Reporting of Observational Studies in Epidemiology (STROBE) reporting guideline.

Clinical data and primary outcome

The research gathered the following patient information: (1) Baseline demographics. (2) Primary causes for ICU admission and clinical comorbidities. (3) Critical care scores: APACHE II and SOFA scores. (4) Laboratory indicators at ICU admission (D1), including markers of infection, indicators of organ function, and lactate. (5) Primary outcome: 28-day mortality rates.

Measurement of serum GDF15 levels

Venous blood specimens were collected using coagulation tubes and then left to rest at room temperature for 30 min. Samples were subjected to centrifugation at 4 °C and 3,000 rpm for 5 min within a 4-h window to isolate the serum. Serum samples from each center were collected and transported to the First Affiliated Hospital of Nanjing Medical University for unified measurement of GDF15 levels, utilizing a human GDF15 enzyme-linked immunosorbent assay kit (Cat# ELH-GDF15: RayBiotech, Guangzhou, China) following the protocol.

Calculation method for GDF15-load

To quantify the GDF15 levels over the first week of ICU stay, we introduced the concept of “GDF15-load” to measure the cumulative change. The formula for calculating GDF15-load is as follows.

GDF15−load=(GDF15D1+GDF15D3)2×(3−1)+(GDF15D3+GDF15D7)2×(7−3).

Statistical analysis

For the development cohort, we employed GBTM to identify changes in serum GDF15 trajectories (Nagin, 2014; van der Nest et al., 2020). The GBTM was constructed by combining model fitting indicators with professional interpretability, which included proportions per class, Akaike information criterion (AIC), Bayesian information criterion (BIC), sample adjusted BIC (SABIC), and relative entropy (Ek).

Data of a continuous nature were depicted as average values ± the standard deviation (SD) when they followed a normal distribution, and as the median accompanied by the interquartile range (IQR) when they deviated from a normal distribution. For comparative analysis between groups, the Student’s t-test was employed for normally distributed data; otherwise, the Mann–Whitney U test or the Kruskal–Wallis test was used. Data of a nominal nature were reported in terms of counts and proportions (%) and examined through the application of the Chi-square test and Fisher’s exact test to determine statistical significance. For comparing subgroups, the analysis of covariance (ANOVA) and the Kruskal Wallis test were utilized, based on the general linear model. Spearman’s correlation test was used to analyze the correlation between GDF15 and other parameters.

Logistic regression analyses, including both univariate and multivariable models, were employed to evaluate the association between candidate predictors and 28-day mortality. For multivariable modeling, variables with statistical significance in univariate logistic regression—including demographics, disease severity scores, baseline laboratory indicators, and GDF15-D1—were included. To assess the independent association between GDF15 trajectories and 28-day mortality, we constructed three hierarchically adjusted multivariable logistic regression models: Model 1 adjusted for demographics; Model 2 further adjusted for disease severity; Model 3 additionally incorporated laboratory parameters reflecting key organ functions.

The predictive ability for clinical outcomes was evaluated by calculating the area under the receiver operating characteristic curve (AUROC), accompanied by 95% confidence intervals (CI). The Youden index was used to calculate the optimal cut-off value for predicting 28-day mortality. Kaplan-Meier curves were plotted to describe differences in clinical outcomes, with the log-rank test applied to assess the significance of differences. The values of Shapley Additive Explanations (SHAP), based on the XGBoost model, were applied to examine the importance of each predictor (Molnar, 2020). For missing values (less than 10%), a single interpolation method was used: mean imputation was employed for data exhibiting normal distribution, otherwise median imputation was utilized.

A p-value below 0.05 was deemed to indicate statistical significance, with all statistical tests being two-sided. The statistical evaluations were conducted utilizing R version 4.4.1 (R Core Team, 2024) and SPSS version 26.0 for analysis.

Results

Clinical characteristics at baseline

A total of 1,973 patients were included in this study, with the enrollment and exclusion process detailed in Fig. 1. Baseline characteristics are detailed in Table S1. The median age was 61 years (IQR 51 to 71 years), with 63.1% (n = 1,244) being male. The median APACHE II score was 18 (IQR 15 to 21), with a mortality rate of 25.8% observed at 28 days. The baseline characteristics between the validation and development cohorts were largely comparable. However, the validation cohort exhibited lower alanine aminotransferase (ALT) and C-reactive protein (CRP) levels, alongside higher serum creatinine (SCr) and activated partial thromboplastin time (APTT) values.

Figure 1 Enrollment and analysis flow of development and validation cohorts.

GDF15 trajectories and derived subtypes in the development cohort

In the development cohort (n = 493), we used GBTM to fit the changes in serum GDF15 levels. The four-class trajectory model was selected for further analysis, demonstrating optimal performance based on AIC, BIC, and SABIC evaluations (Table S2).

Figure 2 illustrates the overall trajectories and the proportion of patients in each of the four subtypes. Based on the distinct patterns of GDF15 changes, we categorized the subtypes as follows: low-maintenance (LM), medium-maintenance (MM), high-increase (HI), and high-persistent (HP). The smallest LM subgroup had consistent GDF15 levels at approximately 2,000 pg/mL during the first week in the ICU. The MM group’s levels were steady at about 4,500 pg/mL. The largest HI subgroup saw a rise from 8,500 pg/mL at admission to 9,700 pg/mL by day 7. The smaller HP group started high at 21,000 pg/mL, dipping slightly to 20,700 pg/mL by day 3 before stabilizing. Detailed trajectories of GDF15 levels are provided in Table 1.

Figure 2 GBTM-derived GDF15 trajectory subtypes and proportions among the development cohort (n = 493).

(A) Classification of ICU patients based on the trajectories of serum GDF15 levels on days 1, 3, and 7 following ICU admission. (B) Proportions of patients in each subtype. LM, Low-maintenance group; MM, medium-maintenance group; HI, high-increase group; HP, high-persistent group.

Table 1 Clinical characteristics and differential analysis across GDF15 subgroups identified by GBTM in ICU patients.

Characteristic	Overall (N = 493)	LM (N = 72)	MM (N = 145)	HI (N = 193)	HP (N = 83)	p-value	
Sex (n, (%))						0.3b	
Male	334 (67.7)	55 (76.4)	99 (68.3)	129 (66.8)	51 (61.4)		
Female	159 (32.3)	17 (23.6)	46 (31.7)	64 (33.2)	32 (38.6)		
Age (years) (Median, (Q1, Q3))	60 (50, 71)	53 (37, 61)	61 (51, 71)	62 (50, 75)	60 (51, 69)	<0.001a,*	
Clinical comorbidities (n, (%))							
Hypertension	119 (24.1)	17 (23.6)	36 (24.8)	48 (24.9)	18 (21.7)	>0.9b	
Diabetes	76 (15.4)	8 (11.1)	24 (16.6)	33 (17.1)	11 (13.3)	0.6b	
Chronic heart failure	98 (19.9)	6 (8.3)	27 (18.6)	50 (25.9)	15 (18.1)	0.013b,*	
Chronic hepatic insufficiency	61 (12.4)	12 (16.7)	14 (9.7)	25 (13.0)	10 (12.0)	0.5b	
Chronic renal insufficiency	111 (22.5)	3 (4.2)	24 (16.6)	59 (30.6)	25 (30.1)	<0.001b,*	
Serum GDF15 levels (pg/mL) (Median, (Q1, Q3))							
GDF15-D1	6,188 (3,287, 11,214)	1,960 (1,503, 2,453)	4,442 (3,281, 5,972)	8,560 (5,948, 11,902)	21,124 (14,167, 25,000)	<0.001a,*	
GDF15-D3	6,033 (3,409, 10,642)	2,237 (1,443, 2,768)	4,621 (3,386, 5,599)	8,830 (5,588, 11,266)	20,638 (11,638, 25,000)	<0.001a,*	
GDF15-D7	6,562 (3,393, 12,121)	2,134 (1,421, 2,600)	4,601 (3,389, 5,818)	9,757 (6,529, 13,644)	20,736 (12,410, 25,000)	<0.001a,*	
GDF15-load	39,167 (22,718, 66,141)	13,359 (9,675, 16,046)	26,341 (21,900, 33,158)	52,997 (41,457, 65,945)	110,356 (89,461, 139,935)	<0.001a,*	
Laboratory tests (Median, (Q1, Q3))							
WBC (×109/L)	12 (8, 16)	12 (8, 16)	11 (8, 14)	12 (9, 16)	12 (9, 16)	0.3a	
PCT (ng/mL)	1 (0, 4)	0 (0, 2)	1 (0, 5)	0 (0, 3)	1 (0, 9)	0.045a,*	
CRP (mg/L)	60 (39, 84)	74 (65, 86)	60 (40, 75)	50 (24, 86)	77 (61, 86)	<0.001a,*	
pro-BNP (pg/mL)	945 (223, 3,903)	579 (126, 2,931)	857 (185, 3,474)	1,239 (237, 3,903)	1,233 (380, 4,210)	0.005a,*	
ALT (U/L)	20 (10, 38)	23 (13, 39)	19 (10, 35)	18 (9, 36)	29 (15, 83)	0.002a,*	
AST (U/L)	33 (21, 61)	31 (21, 58)	32 (20, 54)	33 (21, 58)	45 (27, 120)	0.004a,*	
TBil (μmol/L)	16 (11, 28)	14 (11, 24)	16 (11, 26)	16 (10, 29)	19 (12, 30)	0.2a	
SCr (μmol/L)	85 (61, 141)	63 (51, 88)	77 (61, 103)	103 (69, 176)	113 (69, 206)	<0.001a,*	
BUN (mmol/L)	9 (6, 13)	6 (4, 10)	9 (6, 13)	12 (7, 17)	10 (7, 16)	<0.001a,*	
APTT (s)	32 (28, 38)	30 (27, 34)	32 (28, 38)	33 (29, 39)	33 (28, 40)	0.009a,*	
Lac (mmol/L)	1.6 (1.1, 2.6)	1.2 (0.9, 1.3)	1.5 (1.1, 1.9)	1.8 (1.2, 2.5)	2.6 (2.4, 2.9)	<0.001a,*	
Critical care scores (Median, (Q1, Q3))							
APACHE II	19 (15, 22)	15 (12, 20)	17 (12, 20)	20 (16, 23)	21 (15, 26)	<0.001a,*	
SOFA	8 (5, 10)	5 (4, 6)	6 (4, 8)	9 (8, 10)	10 (6, 12)	<0.001a,*	
28-day mortality (n, (%))	138 (28.0)	8 (11.1)	20 (13.8)	71 (36.8)	39 (47.0)	<0.001b,*	
Notes:

LM, Low-maintenance group; MM, medium-maintenance group; HI, high-increase group; HP, high-persistent group; WBC, white blood cell count; PCT, procalcitonin; CRP, C-reactive protein; pro-BNP, pro-B-type natriuretic peptide; ALT, alanine aminotransferase; AST, aspartate aminotransferase; TBil, total bilirubin; SCr, serum creatinine; BUN: blood urea nitrogen; APTT, activated partial thromboplastin time; Lac, lactate; APACHE II, acute physiology and chronic health evaluation II; SOFA, sequential organ failure assessment.

* p < 0.05, significant statistical difference.

a Kruskal–Wallis test.

b Chi-square test.

Further analysis of the clinical characteristics across the subtypes revealed significant differences in inflammatory markers, organ function, lactate, and critical care scores (Table 1). Patients in the HI and HP groups had higher levels of indicators of organ function, lactate, and critical care scores. Additionally, as initial GDF15 levels increased, the mean levels of aspartate aminotransferase (AST), total bilirubin (TBil), SCr, lactate, and critical care scores also increased, suggesting that patients with elevated GDF15 levels might experience poorer perfusion and organ function.

GDF15 subtypes and clinical outcomes

We evaluated 28-day mortality rates across GDF15 trajectory subtypes in the development cohort (n = 493) to gauge clinical outcome disparities. A marked increase in 28-day mortality rates was seen across GDF15 subtypes LM, MM, HI, and HP, correlating with ascending initial GDF15 levels (Table 1). Additionally, survival analysis indicated notable disparities in overall survival rates among the groups (Fig. 3A). However, further analysis revealed no significant differences in 28-day mortality and survival curves between the LM and MM subgroups, nor between the HI and HP subgroups (Table S3, Figs. 3B and 3C).

Figure 3 Twenty-eight-day Kaplan-Meier survival analysis of the four GDF15 trajectory subtypes in the development cohort (n = 493).

(A) Comparison of 28-day survival curves among the four trajectory subgroups (LM, MM, HI, and HP). (B) Comparison of 28-day survival curves between the LM and MM subgroups. (C) Comparison of 28-day survival curves between the HI and HP subgroups. LM, Low-maintenance group; MM, medium-maintenance group; HI, high-increase group; HP, high-persistent group.

To determine whether these associations were independent of clinical confounders, we conducted multivariable logistic regression with hierarchical adjustments for demographics, disease severity, and laboratory variables (Table 2). After full adjustment, mortality differences remained significant, particularly between the HI/HP and LM subgroups, supporting the independent prognostic value of GDF15 trajectories.

Table 2 Association between serum GDF15 trajectory groups with 28-day mortality.

	LM	MM	p-value	HI	p-value	HP	p-value	
Event/total (%)	8/72 (11.1)	20/145 (13.8)		71/193 (36.8)		39/83 (47.0)		
Unadjusted (OR (95% CI))	Reference	1.3 [0.5–3.1]	0.58	4.7 [2.1–10.3]	<0.01*	7.1 [3.0–16.6]	<0.01*	
Model 1 (OR (95% CI))	Reference	1.1 [0.5–2.8]	0.78	4.2 [1.9–9.3]	<0.01*	6.8 [2.9–16.1]	<0.01*	
Model 2 (OR (95% CI))	Reference	1.1 [0.4–2.8]	0.84	4.6 [1.2–6.8]	<0.01*	4.6 [1.9–11.6]	<0.01*	
Model 3 (OR (95% CI))	Reference	1.0 [0.4–2.6]	0.97	2.8 [1.2–6.7]	<0.01*	4.1 [1.6–10.6]	0.02*	
Notes:

Model 1 was adjusted for baseline age and sex.

Model 2 further adjusted for disease severity using the APACHE II and SOFA score.

Model 3 additionally adjusted for laboratory parameters (PCT, CRP, pro-BNP, ALT, AST, TBil, BUN, Scr, APTT, Lac).

LM, Low-maintenance group; MM, medium-maintenance group; HI, high-increase group; HP, high-persistent group; PCT, procalcitonin; CRP, C-reactive protein; pro-BNP, pro-B-type natriuretic peptide; ALT, alanine aminotransferase; AST, aspartate aminotransferase; TBil, total bilirubin; SCr, serum creatinine; BUN: blood urea nitrogen; APTT, activated partial thromboplastin time; Lac, lactate; APACHE II, acute physiology and chronic health evaluation II; SOFA, sequential organ failure assessment; OR, odds ratio; CI, confidence interval.

* p < 0.05, significant statistical difference.

GDF15-load reflects the cumulative change in GDF15 levels

Analyzing the features of trajectory subtypes, we observed that patients with sustained high GDF15 levels had worse outcomes at 28 days compared to those with lower or moderate levels, regardless of the trend. This suggests that the overall GDF15 level could be a key determinant of 28-day prognosis. We further quantified the cumulative change of GDF15 during the first week of ICU admission using GDF15-load. Significant differences in GDF15-load were observed across the four trajectory subtypes within the development cohort, showing a stepwise rise from the LM to the HP groups (Table 1). We integrated GDF15-load, critical care scores, and other parameters into a SHAP plot to evaluate their relative contributions to the predictive model. Results indicated that GDF15-load was the most significant predictor of 28-day outcomes (Fig. 4). In the receiver operating characteristic (ROC) analysis, predictive power of GDF15-load for 28-day outcomes yielded an AUROC of 0.703, not inferior to both APACHE II and SOFA scores (Fig. 5A). Survival analysis also indicated that individuals with elevated GDF15-load experienced notably reduced 28-day survival probabilities in the development cohort (Fig. S1).

Figure 4 SHAP-determined feature contribution ranking for 28-day mortality in the development cohort (n = 493).

This figure displays the SHAP values, which quantify the impact of each feature on the model’s predictions. The features are ranked according to their contribution to predicting 28-day survival outcomes.

Figure 5 Comparative analysis of GDF15-load and initial GDF15 values (GDF15-D1) in the development cohort (n = 493).

(A) ROC curves for GDF15-D1, GDF15-load, APACHE II score, and SOFA score predicting 28-day outcomes. (B) Correlation between GDF15-D1 and GDF15-load, demonstrating a significant Spearman correlation. (C) Sankey diagram illustrating the agreement between patient groupings based on the cut-off value from the ROC curve of GDF15-D1 and the trajectory-based subgroupings, highlighting the alignment of initial GDF15 levels with the GDF15 trajectory subtypes. LM, Low-maintenance group; MM, medium-maintenance group; HI, high-increase group; HP, high-persistent group; AUROC, area under the receiver operating characteristic curve; CI, confidence interval.

These findings suggest that GDF15-load, derived from the GDF15 trajectory subtypes, serves as a powerful indicator for predicting 28-day outcomes in ICU patients.

Correlation between GDF15-D1 and GDF15-load

Trajectories or GDF15-load require continuous monitoring of GDF15. However, GDF15-D1, also identified as a significant contributor to prognosis in the SHAP plot, offers the advantage of being rapidly and easily available early on. We found a strong correlation of 0.778 between GDF15-D1 and GDF15-load using Spearman’s correlation coefficient in the development cohort (Fig. 5B). We then plotted the ROC curve to evaluate the predictive capacity of GDF15-D1 for 28-day outcomes, which yielded an AUROC of 0.692, still not inferior to both the APACHE II and SOFA scores (Fig. 5A). Based on the ROC curve of GDF15-D1, we calculated the cut-off value (7,582 pg/mL) to divide patients into high and low GDF15-D1 groups. The Sankey diagram showed strong agreement between the GDF15-D1-based groups and the trajectory-based groups (Fig. 5C).

We further validated the predictive ability of GDF15-D1 for 28-day mortality in the validation cohort (n = 1,480). Logistic regression revealed that GDF15-D1 serves as a substantial prognostic factor for 28-day outcomes (odds ratio (OR) = 1.115, 95% CI [1.089–1.141]) (Table S4). The AUROC for GDF15-D1 in predicting 28-day mortality was 0.723 (95% CI [0.692–0.754]), not inferior to the APACHE II score (AUROC = 0.717, 95% CI [0.687–0.747]) (Fig. S2). Survival analysis of the combined cohort (n = 1,973) demonstrated significantly reduced 28-day survival probabilities in individuals with elevated GDF15-D1 levels (Fig. S3).

These findings suggest that serum GDF15-D1 correlates well with GDF15-load and can reliably predict 28-day outcomes in ICU patients, making it a feasible and simplified alternative to the assessment of GDF15 trajectory and GDF15-load.

GDF15-D1 as a valid predictor of 28-day prognosis in postoperative ICU patients

Given the considerable heterogeneity of ICU populations, we then selected the postoperative subgroup with relatively good consistency, to further validate the predictive ability of GDF15-D1 in the validation cohort. This subgroup included 529 patients, with cardiovascular system (34.2%) and brain (24.4%) surgeries being the most common, collectively exhibiting an aggregate mortality rate of 18.3%. Baseline characteristics of these patients are detailed in Table S5. Survivors had a mean GDF15-D1 level of 5,386 pg/mL, contrasted with 12,952 pg/mL in non-survivors, signifying a significant increase in the latter group.

Logistic regression indicated that GDF15-D1 was a significant factor influencing 28-day outcomes in postoperative ICU patients, with an OR of 1.130 (95% CI [1.086–1.174]) (Table S6). The ROC curve for predicting 28-day prognosis also showed that GDF15-D1 had an AUROC of 0.734, showed no significant difference in AUROC between GDF15-D1 and APACHE II (AUROC = 0.728) or SOFA (AUROC = 0.711), indicating comparable predictive ability (Fig. 6A), marking a notable improvement over the heterogeneous ICU population. To further enhance prognostic accuracy, we constructed multivariable models incorporating GDF15-D1 with APACHE II or SOFA. The combined models significantly improved predictive performance (AUROC: GDF15-D1 + APACHE II, 0.816; GDF15-D1 + SOFA, 0.790), both statistically superior to each component alone (all p < 0.05; Fig. 6A). Notably, the integration of GDF15-D1 with both APACHE II and SOFA yielded the highest predictive performance (AUROC = 0.836), suggesting that GDF15-D1 may serve as a valuable complement to established severity scores in postoperative ICU patients. Survival analysis also revealed that individuals with elevated GDF15-D1 had notably reduced 28-day survival probabilities (Fig. 6B).

Figure 6 Evaluation of the prognostic value of GDF15-D1 in postoperative ICU patients (n = 529).

(A) ROC curves and pairwise AUROC comparisons for GDF15-D1, APACHE II, SOFA scores, and their combinations in predicting 28-day outcomes among postoperative ICU patients. (B) Kaplan-Meier survival analysis comparing patients with high vs low levels of GDF15-D1, revealing significantly reduced 28-day survival rates among those with high GDF15-D1 levels. AUROC, Area under the receiver operating characteristic curve; CI, confidence interval.

These findings suggest that serum GDF15-D1 is a strong predictor of 28-day outcomes in postoperative ICU patients, and its integration with existing severity scores can further enhance early risk stratification.

Discussion

This investigation assessed serum GDF15 in critically ill patients across four Chinese ICUs. Analyzing GDF15 at days 1, 3, and 7 post-admission, we identified four distinct trajectory patterns using machine learning. These groups demonstrated substantial variability in baseline characteristics and 28-day mortality rates. We proposed “GDF15-load,” the cumulative GDF15 change over the first 7 ICU days, a potent mortality predictor. Simplifying, we correlated GDF15-load with GDF15-D1, finding a strong link, indicating GDF15-D1’s prognostic significance, especially in postoperative ICU patients, offering a practical and accessible prognostic indicator.

To our knowledge, this is the first study to continuously monitor GDF15 in ICU patients and perform GBTM analysis. Previous studies on GDF15 for prognosis have mainly used single measurements or included it as a risk factor in regression analyses (Hijazi et al., 2018; Pol et al., 2022). However, similar to procalcitonin and lactate, the evaluation of disease severity relies not just on absolute levels but also on the patterns of their fluctuations (Bartoletti et al., 2018; Fuernau et al., 2020). The GBTM is a new machine learning modeling approach that can identify different developmental trajectories within a population and study the associations between these trajectories and predictive factors (Nagin, Jones & Elmer, 2024; Wang et al., 2021b). By applying the GBTM, we identified time-dependent changes in GDF15 trajectories, categorizing patients into four groups. Analysis of clinical characteristics among these groups revealed significant differences in infection markers, organ function indicators, and critical care scores. Survival analysis also demonstrated significant differences in 28-day survival, indicating that continuous monitoring of GDF15 can effectively distinguish critically ill patients with differing severities of disease and clinical characteristics.

What drives outcomes in patients with four GDF15 trajectories? We noted these associated characteristics: 1. Initial GDF15 vary greatly across groups, with LM and MM below average, and HI and HP starting 5–10 times higher, without overlap in later stages. 2. Despite variations in 28-day survival rates, no significant survival difference is seen between LM/MM and HI/HP. 3. Compared to LM/MM, HI and HP have worse organ function and higher critical care scores. Based on these characteristics, we postulate that, beyond the trajectory, the absolute levels of GDF15 also hold considerable significance in assessing the severity of illness and the prognosis. To quantify the overall GDF15 level, we drew inspiration from the concept of “lactate load” in previous studies and introduced the term “GDF15-load” to represent the cumulative amount of GDF15 (Chen, Gong & Yu, 2022; Wu et al., 2024). This concept transformed the trajectory into a continuous numerical variable to facilitate understanding the relationship between GDF15 and patient outcomes. Kruskal Wallis test revealed significant differences in GDF15-load among the groups. SHAP plots and ROC curves demonstrated that the predictive performance of GDF15-load for the 28-day prognosis was not inferior to traditional evaluation systems.

The APACHE score necessitates a 24-h evaluation period and requires a comprehensive collection of clinical data. Whether it is possible to obtain an initial assessment of prognosis upon ICU admission, which could guide treatment, facilitate communication, and establish realistic treatment goals, is also a challenge we aim to address in our study. At present, we have access to point-of-care testing equipment for GDF15, which utilizes fluorescence immunochromatography and can deliver serum GDF15 results within 15 min. Although still in the clinical trial phase, this technology enables rapid measurement at the bedside. Given this context, we further compared the initial GDF15 with the GDF15-load and observed a strong correlation. The initial GDF15 value demonstrated an AUROC for predicting 28-day mortality that was still not inferior to the APACHE score. Thus, we propose that the initial serum GDF15 level at ICU admission serves as a rapid, bedside prognostic biomarker that complements the APACHE score in critically ill patients.

Previous studies have revealed that the accuracy of APACHE II score for prognostic assessment varies among different types of critically ill patients, with higher accuracy for those admitted through the emergency department and lower accuracy for surgical patients (Sungono et al., 2022). Our findings revealed that the AUROC for initial GDF15 is slightly higher than that of the APACHE II score in postoperative ICU patients. The results indicate that, taking into account both rapid simplicity and effectiveness, using the initial GDF15 level for prognosis assessment offers more advantages in the specific population of postoperative patients.

GDF15 is implicated in the activation of macrophages, endothelial dysfunction, and mitochondrial dysfunction (Chen et al., 2024; Chen, Yin & Liu, 2021; Joo et al., 2023; Rochette et al., 2020; Xu et al., 2022). Normally expressed at low levels, GDF15 is markedly upregulated under stress such as inflammation, hypoxia, and tissue damage (Wang et al., 2021a). Moreover, recent studies have delved into the protective role of GDF15 in a variety of diseases (Li et al., 2023, 2024; Lu et al., 2024). As a multifunctional cytokine, GDF15 plays a crucial role in the onset and progression of critical illness, suggesting it as a potential target for diagnosis and treatment, and providing new opportunities for clinical intervention.

This study has several limitations. First, it focused on the association between GDF15 and 28-day prognosis, neglecting longer-term prognosis and other clinical outcomes. Second, although multicenter, the number of patients with complete longitudinal GDF15 data was limited, restricting disease-specific subgroup analyses. Third, GDF15 was measured only on days 1, 3, and 7, missing intermediate or later time points and limiting the resolution of trajectory modeling. Fourth, all samples were analyzed using the RayBiotech ELISA, which has not been benchmarked against the clinically used Roche Elecsys assay, limiting cross-platform comparability. Fifth, the assay’s ability to detect the common H202D variant has not been validated, which may impact absolute GDF15 levels in DD homozygotes, although internal consistency across the cohort was maintained. Future studies should include larger and more diverse cohorts, more frequent sampling, and cross-platform assay validation—including genetic variant sensitivity—to facilitate clinical translation.

Conclusions

Serum GDF15 trajectories are valuable predictors of 28-day mortality in critically ill patients. The initial GDF15 level at ICU admission (GDF15-D1) is a practical and effective early prognostic marker, particularly in postoperative patients, as it strongly correlates with both GDF15 trajectories and GDF15-load. When available, serial GDF15 monitoring and the calculation of GDF15-load further improve predictive accuracy. These findings underscore the potential of GDF15 as both an early and dynamic biomarker for risk stratification in the ICU.

Supplemental Information

Supplemental Information 1 28-day survival analysis in the development cohort (n = 493) by high/low GDF15-load.

Supplemental Information 2 ROC curves for GDF15-D1, APACHE II and SOFA scores in the validation cohort predicting 28-day outcomes (n = 1,480).

Supplemental Information 3 28-day survival analysis in the combined cohort (n = 1,973) by high/low GDF15-load.

Supplemental Information 4 Baseline characteristics of study participants.

Supplemental Information 5 Model fit evaluation for GBTM-based serum GDF15 level trajectory analysis in ICU patients.

Supplemental Information 6 Comparison of 28-day mortality rates among GDF15 trajectory subtypes.

Supplemental Information 7 Multivariate logistic regression model of predictors for 28-day mortality in the validation cohort.

Supplemental Information 8 Clinical characteristics and differential analysis of postoperative ICU patient in the validation cohort.

Supplemental Information 9 Multivariate logistic regression model of predictors for 28-day mortality in postoperative ICU patients.

Supplemental Information 10 STROBE checklist.

Supplemental Information 11 Data.

Supplemental Information 12 Codebook.

Supplemental Information 13 Code-GBTM.

Supplemental Information 14 Code-xgboost-shap.

We would like to extend our sincere appreciation to Dr. Ni Shi from the Second Affiliated Hospital of Harbin Medical University, Dr. Baohua Zhu from Nanjing Central Hospital, Dr. Jiaqiong Li from Xuzhou Central Hospital, and Dr. Bo Yun from the First Affiliated Hospital of Nanjing Medical University for their invaluable assistance in patient enrollment and data collection for this study. We are also deeply grateful to Professor Qiulun Lu and Professor Yan Chen for their invaluable support in the study’s design and execution.

Additional Information and Declarations

Competing Interests

The authors declare that they have no competing interests.

Author Contributions

Qinxue Wang conceived and designed the experiments, performed the experiments, analyzed the data, prepared figures and/or tables, authored or reviewed drafts of the article, and approved the final draft.

Jiawei Wang conceived and designed the experiments, performed the experiments, analyzed the data, prepared figures and/or tables, authored or reviewed drafts of the article, and approved the final draft.

Yuhan Zhao performed the experiments, analyzed the data, prepared figures and/or tables, authored or reviewed drafts of the article, and approved the final draft.

Yuanze Ma performed the experiments, prepared figures and/or tables, and approved the final draft.

Xiang Li performed the experiments, prepared figures and/or tables, and approved the final draft.

Xinyi Chang performed the experiments, prepared figures and/or tables, and approved the final draft.

Nan Zheng performed the experiments, prepared figures and/or tables, and approved the final draft.

Yong Ji conceived and designed the experiments, authored or reviewed drafts of the article, and approved the final draft.

Yi Han conceived and designed the experiments, authored or reviewed drafts of the article, and approved the final draft.

Human Ethics

The following information was supplied relating to ethical approvals (i.e., approving body and any reference numbers):

This study have been approved by the Institutional Study Board of the First Affiliated Hospital of Nanjing Medical University (2022-SR-678) and the Second Affiliated Hospital of Harbin Medical University (KY2024-066-01).

Data Availability

The following information was supplied regarding data availability:

The raw data and code are in the Supplemental Files.

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
