# Peer review of "Serum growth differentiation factor 15 trajectory predicts 28-day mortality in critically ill patients: a multicenter cohort study"

_PeerJ, doi:10.7717/peerj.20317_

## Round 0.1 · original submission · Major Revisions

· Academic Editor

Major Revisions

**Language Note:** When you prepare your next revision, please either (i) have a colleague who is proficient in English and familiar with the subject matter review your manuscript, or (ii) contact a professional editing service to review your manuscript. PeerJ can provide language editing services - you can contact us at [email protected] for pricing (be sure to provide your manuscript number and title). – PeerJ Staff

·

Basic reporting

This manuscript describes a multicenter cohort study investigating the predictive value of serum GDF-15 levels for 28-day mortality in 1,973 critically ill patients. Of these, 493 comprised the cohort for development with full serum profiles on days 1, 3, and 7. The study describes different GDF-15 progression patterns on days 1, 3, and 7 (low (LM), moderate (MM), high rising (HI), and high falling (HD)) that correlate with 28-day mortality. Mortality was highest in the HI and HD groups. To quantify the cumulative change in GDF-15 levels over the first seven days in the ICU, the study introduces the term “GDF15 load”, which shows a significant increase from LM to HD. GDF-15 load was the strongest predictor of 28-day mortality, with an AUROC of 0.703. It was non-inferior to the Acute Physiology and Chronic Health Evaluation (APACHE II) and Sequential Organ Failure Assessment (SOFA) scores.

The initial GDF-15 value proved to be a rapid and practical prognostic marker, especially in postoperative patients. A strong correlation of 0.778 between GDF-15-D1 and GDF-15 load was observed. The AUROC for GDF-15-D1 for predicting 28-day mortality was 0.692, making it a useful indicator. In the postoperative cohort, the AUROC for GDF-15-D1 was 0.734, which represents an improvement over the traditional scores. Moreover, GDF-15-D1 values are much easier and faster to assess than these much more complex scores.
The report is written in clear English and conforms to the standards of reporting scientific findings in the field. The introduction is concise, but adequate. The structure conforms to discipline norms. References are appropriate.

Experimental design

The research question is well defined and likely relevant for doctors treating critically ill patients. Methods are clearly described and suitable (criticism see below). Ethical approval and informed consent were obtained. The figures are convincing. Raw data are supplied. The discussion is fine. The conclusion is also correct (see comment below).

Validity of the findings

Altogether, the findings appear valid. Still, I have the following criticisms:

1. Serum GDF-15 trajectory and GDF-15-load are interesting concepts. For practical reasons, however, early clinical decisions could only be based on initial GDF-15 serum levels at ICU admission. Thus, I suggest emphasizing the findings on the relevance of initial GDF-15 in the conclusion, and mention that longitudinal serum GDF-15 trajectories and GDF-15 load further increase the accuracy of prediction.

2. Was a training and a validation set included? If not, please explain why!

3. Figures 3 and S1 are based on 493 patients. Could a Kaplan-Meier plot be generated from the GDF-15-D1 values of all 1,973 patients? Throughout the manuscript, it should be indicated more clearly where data are based on the whole cohort of 1,973 patients, or the development cohort (training set?) of 493 patients.

4. The human GDF15 enzyme-linked immunosorbent assay kit (RayBiotech, China, Cat# ELH-GDF15) was used for the study. In most clinical settings, the Roche Elecsys GDF-15 assay would likely be used. Do you have any benchmarking data to show that these assays give similar results? For any clinical application, it would be important to make sure the choice of assay does not affect the results.

5. What is known about the recognition of the GDF15 H202D variant by the used ELISA? (compare Y Karusheva et al, The Common H202D Variant in GDF-15 Does Not Affect Its Bioactivity but Can Significantly Interfere with Measurement of Its Circulating Levels, The Journal of Applied Laboratory Medicine, Volume 7, Issue 6, November 2022, Pages 1388–1400, https://doi.org/10.1093/jalm/jfac055)

Additional comments

1. The name and definition of the high-falling (HD) group are misleading, as the drop in GDF-15 serum levels is only minimal. Figure 2 shows that patients in this group have the highest GDF-15 levels over the whole course of the study, far higher than patients in the high rising group. (Patients showing a massive decrease in GDF-15 levels might have a different outcome than patients with largely maintained very high levels.) Please provide a clear definition of this group and think about finding an alternative name (high, initial peak; high, no further increase; or other)

2. Regarding GDF-15, it should be mentioned that GDF-15 is commonly overexpressed in cancers. Therefore, it is questionable whether the proposed score would still work in patients with malignancy. Were such patients excluded?

Altogether, this is an interesting manuscript. However, GDF-15 is a tricky molecule. Thus, it is important to sort out potential technical issues caused by the chosen ELISA assay. Moreover, it is difficult to understand patient selection, as the overall number of patients included in the study far exceeded the number of patients analyzed in the key figures.

·

Basic reporting

Language: English is satisfactory overall, though the introduction could benefit from improvements in clarity.

Line 62–64: “However, the APACHE scoring system also has its limitations. It involves a large number of physiological and laboratory parameters, leading to complexities and delays in clinical application.” While APACHE includes many parameters, most are low-cost, routinely collected, and often automatically calculated in the EHR. The authors should consider acknowledging this.

Research gap: The introduction does not clearly identify the research gap. While it discusses GDF15 in general terms, it omits reference to several important prior studies on GDF15 in critical illness over the past decade (e.g., PMID: 29180833, 30633545, 23706007, 31310811, 18263797, 31076013). The authors should frame the introduction more explicitly around critical care and articulate how their study adds to or differs from previous research. Why is “GDF15-load” important, and what new insights do the authors hope to provide? Currently, the aims are listed at the end of the introduction without justification. This style is more suitable for the methods section. Instead, the authors should explain the rationale behind their goals in the introduction in more general terms.

Experimental design

Inclusion/exclusion: Line 95: “One or more organ failures” is vague—please specify the types of organ failure being considered and ensure this is clearly outlined in the protocol (consider attaching it as supplementary material). The statement “ICU admission within ≤24 hours and ICU duration ≥48 hours” is confusing. If the intent is to include patients admitted within 24 hours and expected to stay in the ICU for at least 48 hours, this should be clearly stated. Otherwise, define the criteria more precisely. The exclusion criterion “hospital stay >7 days before ICU admission” is unclear. Please justify this decision—why is prolonged hospitalization before ICU admission a reason for exclusion?

Statistical analysis: The authors are commended for their robust statistical approach.

Results/methods: The explanation of the GDF15-load formula would be more appropriate in the methods section, rather than being introduced in the results.

Validity of the findings

-

Additional comments

Discussion: The discussion would benefit from commentary on the cost and practicality of using GDF15 as a biomarker in routine clinical practice. Additionally, addressing why it has not yet been widely adopted would strengthen the context and relevance of the findings.

Reviewer 3 ·

Basic reporting

The article describes valuable scientific knowledge and is written in a clear and objective manner with fluid English. This study investigates the predictive utility of serum GDF15 trajectories for 28-day mortality among patients in the intensive care unit (ICU).

The Abstract has a correct and organised structure; however, I think it would be useful to briefly describe the statistical modelling methodology to analyse the data.

The statistical analysis methods seems to be appropriate to the goals of this work, however to conclude about predictive ability of serum GDF15 trajectories for 28-day mortality, would be more robust if the model were adjusted for clinical and laboratory variables, including the Charlson Comorbidity Index, to classify comorbidity conditions which may influence mortality risk.

In lines 121 - 125, it makes sense to add a reference that explains the Group-Based Trajectory Model (GBTM).

Suggestions: Nagin, D. S. (2014). Group-based trajectory modelling: An overview. Annals of Nutrition and Metabolism, 65(2–3), 205–210. Karger Publishers. https://doi.org/10.1159/000360229
or
van der Nest, G., Lima Passos, V., Candel, M. J. J. M., & van Breukelen, G. J. P. (2020). An overview of mixture modelling for latent evolutions in longitudinal data: Modelling approaches, fit statistics and software. Advances in Life Course Research, 43(January), 100323. https://doi.org/10.1016/j.alcr.2019.100323

Again, in Statistical Analysis, lines 141, 142, the use of Shapley Additive Explanations (SHAP) needs a reference to clarify the methodology of this method. For example, Christoph M. Interpretable machine learning: A guide for making black box models explainable (Leanpub). 2020.

Experimental design

Study design and population

The study design and Population were well defined, as well as the research question.

GDF15 levels were measured only at days 1, 3, and 7, which I consider a few measures to build the trajectory patterns.

In lines 94, 95 please could you please clarify the Inclusion criteria: 2. <= 24 hours of ICU admission?

Line 155, please could you clarify how you compare characteristics between the validation and development cohorts?

On lines 219 and 220, you used logistic regression, but you reported (hazard ratio estimate [HR] = 1.115, which is based on the estimation of a survival analysis model. Could you clarify this point, please? The same about the results in tables S4 and S6.

Validity of the findings

In lines 229 and 230 you mentioned that was selected the postoperative subgroup with relatively good consistency, to further validate the predictive ability of GDF15- D1 in the validation cohort. Which criterium did you use to choose this subgroup of 529 patients. which measures did you evaluate to select this group?

In Figure 6 you show the discriminative ability of APACHE II, SOFA, and GDF15-D1, to conclude by the superiority of GDF15-D1, you should present the tests to compare the AUC's. May be they don't differ significantly. Moreover, it would be interesting to calculate the AUC of the multivariable adjusted model, not only

In the legend of Table 1, table S1, i recommend to include a symbol to identify which test originate the reported p-value.

In Table S4, you again report HR, and in the title you have "Multivariate logistic regression model ". Could you clarify which model was adjusted? These results are from multivariable model? If yes, which methods did you used to selected the variables?

Conclusions are well stated, and the strengths and limitations are also well described.

---

## Round 0.2 · accepted · Accept

· Academic Editor

Accept

Thank you for revising your manuscript to address the concerns of the reviewers. Reviewers 2 and 3 now recommend acceptance and I am satisfied that the comments of reviewer 1 have been addressed. The manuscript is now ready for publication.

·

Basic reporting

Responded and addressed all comments appropriately. Performed additional analysis which is commendable. Recommend publication.

Experimental design

All good now

Validity of the findings

All good now

Additional comments

No

Reviewer 3 ·

Basic reporting

The authors addressed the questions and suggestions raised in the review and made any possible changes. Therefore, I consider the article to be ready for publication.

Experimental design

I have no comments.

Validity of the findings

I have no comments.

Additional comments

I have no comments.